# N, S Dual-Doped Carbon Derived from Dye Sludge by Using Polymeric Flocculant as Soft Template

**DOI:** 10.3390/nano9070991

**Published:** 2019-07-09

**Authors:** Daofeng Luan, Liang Wu, Tingting Wei, Liu Liu, Yin Lv, Feng Yu, Long Chen, Yulin Shi

**Affiliations:** Key Laboratory for Green Processing of Chemical Engineering of Xinjiang Bingtuan, School of Chemistry and Chemical Engineering, Shihezi University, Shihezi 832003, China

**Keywords:** dye sludge, flocculant, soft template, N, S dual-doped, oxygen reduction reaction

## Abstract

Dye sludge is a major by-product and it will bring critical environmental problems in the textile industry. In this study, dicyandiamide formaldehyde resin (DFR) is used as an effective flocculating agent for the removal of anionic dyes from textile dye wastewater. Employing dye-contaminated sewage sludges as precursors, N, S dual-doped carbon materials are successfully synthesized by using DFR as a soft template. The specific surface area, morphology, and pore structure of the resulting annealed products can be easily controlled by changing the DFR content of the dye sludge. The oxygen reduction reaction performance of optimal carbon material (N, S-DF-2) is close to commercial 20% Pt/C in alkaline medium, including onset potential (0.98 V), half-wave potential (0.82 V), as well as limiting current density (5.46 mA·cm^−2^). Furthermore, it also shows better durability and crossover resistance. In addition, N, S-DF-2 exhibits a large specific capacitance (230 F·g^−1^ at 1 A·g^−1^) and super capacitance retention (nearly 98% at 10 A·g^−1^) after 2500 cycles as supercapacitors electrodes. This work opens up a new method to take full advantage of organic polymeric flocculant as a soft template to prepare N, S dual-doped carbon materials, which will be beneficial for the reuse and recycling of sewage sludge, as well as for the production of good quality energy conversion and storage materials.

## 1. Introduction

It is estimated that over 8 × 10^5^ tons of commercial dyes and pigments are applied in the textile industry annually [1,2]. Azo dyes make up more than one-half of all dyes. Approximately 10%–15% of azo dyes do not bind to the textile fibers in the dyeing process and are released into the environment in the dyeing process [3,4]. Normally, azo dyes based on the azo structure with aromatic backbone carry one or more sulfonic acid groups (-SO_3_H). Therefore, azo dyes are easily soluble in water leading to an undesirable color, eventually causing major damage to the aquatic environment due to their high resistance to natural degradation. 

Several treatment technologies, such as adsorption, chemical oxidation, photo degradation, and even biological degradation, have been used to treat high-concentration dye wastewater extensively. Among them, coagulation–flocculation and adsorption are considered effective methods. However, the two methods only convert azo dye wastewater into huge solid wastes, which raises issues and challenges for further disposal of these hazardous sewage sludge (SS). Transforming environmental wastes to utilizable energy materials is a promising strategy for economic sustainable development considering that it concerns environment and energy aspects [5,6].

As emerging green energy conversion and storage devices, fuel cells and supercapacitors (SCs) attract attention of researchers [7]. Efficient catalysts are the essential materials for fuel cells. However, their development and application have been limited in the cathode sluggish kinetics [8]. In addition, Pt-based catalysts find difficulties in commercialization because of scarcity, expensiveness and poor stability [9]. To deal with the challenge, recent researches pay more attention to design a series of non-precious electrocatalysts to replace expensive Pt/C [10]. Carbon-based material is a promising candidate for electrochemical energy conversion and storage fields, including electrical double-layer capacitors (EDLCs), supercapacitors (SCs), and oxygen reduction reaction (ORR) catalysts [11]. 

To develop excellent electrochemical performance of carbon materials, heteroatom doping has attracted increasing attention because of higher surface wettability, smaller charge transfer resistance, and charge-storage capability [12,13]. N, S dual-doped carbon materials, including carbon nanosheets [14,15], 3D honeycomb-like carbon, and biomass-derived carbon [13,16], have been widely focused on energy application (ORR and SCs). Further studies also indicated that N, S dual-doped carbon materials could offer multiple benefits and synergistic effects, thereby enhancing electrochemical performance of the energy devices [17,18]. However, the main disadvantages associated with N, S dual-doped carbon materials are high-cost, tedious preparation process, and the use of toxic chemicals, which significantly hinder their further development in electrical energy storage technologies [19,20]. Searching for abundant and low-cost resources as precursors to prepare N, S dual-doped carbon materials attracts large interest. Flocculation sludge represents an attractive candidate as precursors to produce N, S-doped carbon materials, because it contains abundant N and S atoms inherently. 

In this paper, we develop a new soft-template method for synthesizing N, S dual-doped carbon materials (N, S-DF-x) by using dye flocculation sludge as precursor (Scheme 1). For the first time, dicyandiamide formaldehyde resin (DFR) is exploited as both a flocculant and soft-template to synthesize N, S dual-doped carbon materials. Dye sludge were prepared by coagulation–flocculation method using DFR as flocculant. A variety of N, S dual-doped carbon material could be readily prepared by flocculant (template) removal during calcination. To evaluate the influence of flocculant on the pyrolysis product, various sewage sludge with different DFR content were taken as precursor for the preparation of N, S dual-doped carbon materials. The N, S dual-doped carbon materials with optimal DFR content deliver remarkable electrochemical properties of ORR and SCs, such as positive onsets potentials (0.98 V), half-ware potentials (0.82 V), and energy storage performance (230.5 F·g^−1^ at 1 A·g^−1^) in alkaline media. We believe this green process can achieve re-use of hazardous dye sludge as well as design a pioneering thought for environmental protection and development of sustainable energy strategies.

## 2. Materials and Methods 

### 2.1. Materials and Reagents

Dicyandiamide formaldehyde resin (DFR) were obtained from XinJiang RuYi textile and garment Co., Ltd. Azo dye reactive brilliant red (K-2BP, λ_max_ = 534.5 nm), a widely used commercial dye, was obtained from Longsheng Dyestuff Co., Ltd. Zinc chloride, sodium hydroxide (98% pure), and potassium acetate (99% pure) were obtained from Shengao Chemical Co., Ltd (Tianjin, China). Hydrochloric acid (38% pure) was gained from Lvjian chemical Co., Ltd (Shanghai, China). All other chemicals were analytical grade. Deionized water was used in all the experiments.

### 2.2. Azo Dye Flocculation and Sludge Flocs Preparation

Through the hydrolysis process, K-2BP is transformed into HK2BP, being an actual component of dye wastewater according to our previous work [21]. HK2BP (100 mg·L^−1^, 180 mL) was average transferred to nine beakers and DFR (50 mg·L^−1^, 2–10 mL) was added, respectively. Finally, proper amount of deionized water was added and the total volume was constant at 80 mL. pH was adjusted to 8.0 through 1 M HCl or NaOH. The standard experiments steps were: (1) Stirring for 10 min at 100 rpm; (2) finally, the flocs were obtained by standing at 1 h. The supernatant was obtained by filtration. The concentration of dye residual liquid was measured by ultraviolet visible spectrophotometer at λ_max_ = 534.5 nm. The color removal (*R%*) and dye flocculation capacity *q* (mg·g^−1^) were calculated as follows:(1)$$R\%=\frac{C_{0}-C_{i}}{C_{0}}\times 100$$
(2)$$q=\frac{V\left(C_{0}-C_{i}\right)}{m}$$ where *C*_0_ and *C_i_* are the dye concentration before and after flocculation (g·L^−1^), respectively. *V* is the volume of the dye solution (L), and *m* is the weight of the DFR (material, g).

### 2.3. Synthesis of N, S Dual-Doped Carbon Materials N, S-DF-x

Flocculation sludge pyrolysis is divided into two steps: Pre-carbonization of sludge and the activation of zinc chloride. Sludge flocs were prepared in the optimum flocculation conditions (R = 97.3%, q = 8000 mg·g^−1^). The sludge flocs were filtered to obtain the cake for freeze-drying, and then the dried flocs were used for grinding. To evaluate the influence of flocculant on the pyrolysis product, various sewage sludge with different DFR content were taken as precursor for the preparation of N, S dual-doped carbon materials. Typically, 6.0 g HK2BP and 1.0 g flocs (the content of DFR is 1.6%) were mixed uniformly and then transferred into pipe furnace. The reaction heated to 500 °C for 2 h with a ramp rate of 5 °C·min^−1^ under high purity argon (99.999%, 120 cm^3^·min^−1^) atmosphere. After the temperature cooled down, the obtained black powders were washed with 1 M HCl, then distilled water up to neutrality, aiming to remove impurities in carbon materials. Finally, pre-carbonization products for template removal were obtained after being dried at 80 °C overnight. 

In the second step, pre-carbonization products were mixed with ZnCl_2_ at a mass ratio of 1:3, then dissolved in deionized water and stirred for 24 h. Then, the mixture was dried by rotary evaporator and transferred into pipe furnace for heating treatment. The reaction was heated to 300 °C for 1 h and heated to 800 °C for 2 h with a ramp rate of 5 °C·min^−1^ under argon atmosphere. The product was washed and dried in an oven in the same conditions as above mentioned. Finally, N, S dual-doped carbon materials (the content of DFR are 1.0%, 1.6%, and 2.9% corresponding to N, S-DF-x (x = 1, 2, 3)) were obtained. 

### 2.4. Materials Characterization

Morphologies, structures, and surface composition of the as-prepared N, S-DF-x (x = 1, 2, 3) samples were characterized by X-ray diffraction (XRD, Bruker D8, Zeiss, Karlsruhe, German) with Cu Kα radiation (α = 1.54059 nm); transmission electron microscopy (TEM, JEM-2100F, JEOL, Tokyo, Japan), with acceleration voltage at 300 kV; scanning electron microscopy (SEM, Gemini-500, Zeiss, German), equipped with an energy-dispersive spectrometer (EDS, Zeiss, German); and X-ray photoelectron spectroscopy (XPS, ESCALAB 250, Thermo Fisher Scientific Company, Waltham, MA, USA). Electron microscope operated at an acceleration voltage of 300 and 50 kV corresponding to TEM and SEM, respectively. XPS test with an Al (mono) X-ray source and the binding energies of the carbon materials were calibrated according to adventitious carbon (binding energy = 284.6 eV). Raman spectra of N, S-DF-x carbon materials powders were obtained from a RM-2000 Raman spectrometer (Renishaw Invia, Renishaw, Wotton-under-Edge, Gloucestershire, UK) by a 532 nm laser excitation. Brunauer–Emmett–Teller (BET) specific surface area and pore properties of N, S-DF-x were conducted on a JW-BK132F (JWGB, Beijing, China) system at liquid-N_2_ temperature.

### 2.5. Electrochemical Characterization

#### 2.5.1. ORR Measurements

A CHI760E electrochemical workstation (Chenhua, Shanghai, China) was used to investigate the electrocatalytic property of the N, S-DF-x samples. N, S-DF-x samples (3 mg) were put into ethanol (1 mL), sonicated for 30 min by sonication, and finally the catalytic ink was obtained. Next, 10 μL of the N, S-DF-x was loaded on a glassy carbon (GC) electrode (3 mm in diameter). On the GC electrode, a thin layer of powder could clearly be seen. The GC electrode was fixed on a rotating disk electrode (RDE) as the working electrode (WE). Pt foil (1 cm^2^) served as counter electrode (CE), so the reference electrode (RE) was Hg/Hg_2_Cl_2_. ORR test was carried out in a N_2_/O_2_-saturated 0.1 M KOH electrolyte. The rotation speed was controlled at 400 to 2025 rpm.

The results of rotation disk electrode (RDE) experiments were obtained by the following equations:
(3)$$\frac{1}{J}=\frac{1}{J_{L}}+\frac{1}{J_{K}}=\frac{1}{B\omega^{\frac{1}{2}}}+\frac{1}{J_{K}}$$ where *J* is the measuring current density, *J_L_* represent diffusion-limiting current densities, *J_k_* represent kinetic-limiting current density, *ω* is rate of electrode rotating, and *B* can be figured out by slope of K–L plot. 

(4)$$B=0.2\mathrm{nF}C_{0}D_{0}^{\frac{2}{3}}\;\nu^{-\frac{1}{6}}$$ where *n* is the transferred electron number, Faraday constant (F = 96,485 C), bulk concentration (*C*_0_ = 1.2 × 10^−3^ mol·cm^−3^) of O_2_, kinematic viscosity (*ν* = 1.1 × 10^−2^ cm^2^·s^−1^) of the electrolyte, and oxygen diffusion coefficient (*D*_0_ = 1.9 × 10^−5^ cm^2^·s^−1^).

Electron transfer number (n) and H_2_O_2_ yield was obtained by ring-rotated disk electrode (RRDE) test. Specific calculation by the Equations (5) and (6):(5)$$n=4\times \frac{I_{D}}{I_{D}+\frac{I_{R}}{N}}$$
(6)$$H_{2}O_{2}\%=\frac{\frac{I_{R}}{N}}{I_{D}+\frac{I_{R}}{N}}$$

Here, ring current (*I_R_*), disk current (*I_D_*), and collection efficiency of the Pt ring (*N* = 0.37) can be measured precisely.

#### 2.5.2. Capacitive Measurements

Electrochemical tests were evaluated on CHI760E electrochemical workstation (Chenhua, Shanghai, China). N, S-DF-x active materials, PTFE, and acetylene black were mixed together, then coated on the nickel foam as working electrodes. This is a three-electrode system (6 M KOH electrolyte), whereby N, S-DF-x active materials on the working electrode approximate 3 mg·cm^−2^. Cyclic voltammetry (CV), galvanostatic charge–discharge (GCD), and cycle-life stability of N, S-DF-x were measured with counter electrode (Platinum foil) and reference electrode (Hg/HgO).

The specific capacitance of GCD was calculated by equation:
(7)$$C_{s}=\frac{I\Delta C}{m\Delta V}$$ where *I* is current (A), *m* represents the mass (g) of N, S-DF-x, *Δt* refers to time (s), and *ΔV* is the voltage change (V) during the discharge process.

## 3. Results and Discussion

### 3.1. Preparation and Composition of Flocculation Sludge

We explored the effect of the DFR flocculant dosage on dye HK2BP, various dosages (4–28 mg·L^−1^) were exposed to a fixed dye concentration (100 mg·L^−1^) at pH 8.0. Generally, a low flocculant dosage with simultaneously high dye removal efficiency is desirable for industrial dye wastewater treatment. The effects of the dose on the removal of dye HK2BP are shown in Figure 1a. The results signified that dye removal increased with the increasing dose of flocculant, reached a maximum at the optimal concentration of about 12 mg·L^−1^, and then decreased with a further increase in dose. According to Equations (1) and (2), when the dye removal rate reaches 97.3%, the adsorption capacity is 8000 mg·g^−1^. Moreover, the commercial coagulants DFR displays color removal results evidently at a wider range of pH values (Figure 1b). Dye sludge flocs are actually made up of DFR flocculant and dye HK2BP, which account for 11% and 89%, respectively. Inherent N and S atoms originate from dye HK2BP (Figure 1c), the significant characteristic enables dye sludge to be a promising precursor for synthesizing N, S dual-doped carbon materials. So, a series of N, S dual-doped carbon materials (Figure 1d) were prepared from dye-containing sludge with different DFR content. 

### 3.2. Characterization of N, S-DF-x.

Porous carbons materials can be easily synthesized by the soft-template method at low carbonization temperatures [22]. Thus, the soft-template approach to prepare N, S-DF-x porous carbon can make pre-carbonization products ideal candidates for electrochemical materials. Concretely, infiltration of an appropriate soft-template DFR by pre-carbonization products, followed by template removal in a high-temperature carbonization process [23]. To explore the thermal stability of flocculation sludge ingredients, thermogravimetry (TG) analysis was performed in argon gas at a heating rate of 10 °C·min^−1^. HK2BP displayed better thermal stability than DFR, though they all started to lose weight at 200 °C evidently (Figure 2a). Moreover, the residual mass of DFR was far below 5% when the temperature reached 800 °C. From Figure 2b, two obvious peaks of DFR corresponded to large weight loss in the pyrolysis process while the derivative thermogravimetry (DTG) curve of HK2BP was relatively flat. Thus, DFR as a soft template was removed basically by high-temperature carbonization. Further studies were performed using N_2_ adsorption-desorption isotherms. According to Figure 2c, all samples exhibited typical IV curves with H_3_-type hysteresis loops [24]. Furthermore, at lower relative pressure (P/P_0_ < 0.4), the adsorption sharply increased. This shows that the carbon materials had more micropores. According to the trend of curves, N, S-DF-2 had more abundant micropores than N, S-DF-1 and N, S-DF-3. Moreover, N, S-DF-2 materials possessed obvious mesoporous because the curve had a distinct hysteresis loop. Pore size distribution curves (Figure 2d) clearly show the micropore and mesopore distribution of all samples. But the number of micropores and mesopores of N, S-DF-2 was superior to N, S-DF-x (x = 1, 3) carbon materials (ESI Appendix A). Besides, N, S-DF-2 materials possessed a larger specific surface area (801.14 m^2^·g^−1^) than N, S-DF-x (x = 1, 3); which also corresponds to the porous structures of N, S-DF-2 materials. The results demonstrated a dual role of DFR as both a flocculant and pore-forming agent [25]. The moderate doping amount of DFR contributes to the formation of abundant micropores and mesopores. However, when the DFR ratio increased, carbon nanostructures totally collapsed after removing the DFR template [23]. Thus, compared with N, S-DF-2, the number of internal pores and specific surface area declined obviously in N, S-DF-3 carbon materials (ESI Appendix A). However, compared with N, S-DF-1, the appropriate content of DFR in N, S-DF-2 mad the carbon materials possess porous structures and a large specific surface area. Notably, carbon materials’ electrochemical properties were influenced by porous structure and specific surface area [26]. Mesoporous structures aid the effect of electrolyte ions and reactants in the oxygen reduction reaction [27]. For most electrode materials, particularly, mesopores also can relieve volume expansion by providing more void space effectively during the electrochemical redox [28]. The large specific surface area and abundant micropore structures enhance electrochemical energy storage performances [29]. Therefore, the as-synthesized N, S-DF-2 carbon materials are expected to exhibit promising utilization properties for supercapacitors and ORR.

The SEM images of the N, S-DF-1 and N, S-DF-2 catalysts carbonized under 800 °C are shown in Figure 3a,b. N, S-DF-1 displayed small roughness and low porosity on the surface. This may be due to the soft-template DFR being too little to produce enough gas-bubbling during the pyrolysis process. Compared with N, S-DF-1, N, S-DF-2 had a high rough surface with a large number of unevenly-distributed holes. The structures exposed ample contact surface area for electrochemical activities and electrolytes. Additionally, carbon materials’ porosity facilitate the adsorption and diffusion of ions and reactants of ORR [30]. A hierarchical porous network is expected to facilitate the adsorption and diffusion of ions and reactants, thus improving the ORR activity [31]. But N, S-DF-3 (ESI Appendix A) showed less pore structures than N, S-DF-2, because the carbon skeleton collapsed badly after the carbonization process and the DFR content increased. Moreover, the TEM image of N, S-DF-2 displayed a sheet-like porous structure (Figure 3d). Observed at close range, some wrinkles on the gauzy lamella could be seen. But the nanosheets of N, S-DF-1 and N, S-DF-3 were thick relatively (Figure 3c, ESI Appendix A). Through attentive observation, the pyrolysis process of the samples created defects and wrinkles on N, S-DF-2 carbon sheets, and plentiful pores were formed among the wrinkles accordingly. The remarkable feature could provide more active sites for electrolyte ions as well as enhance the capacitance of electrode materials [32,33]. The elemental SEM-mapping photograph of Figure 3f,g reveal the uniform distribution of N, S, in the N, S-DF-2 carbon materials skeleton. Figure 3h shows the energy dispersive spectroscopy (EDS) of the N, S-DF-2 sample, and the content of N (10.2%) and S (2.2%) elements were higher. In addition, the content of N in N, S-DF-2 carbon materials were maximum and far more than N, S-DF-1 and N, S-DF-3 (ESI Appendix A). The content of N increased, and not only promoted electron transfer in ORR but also optimized the electronic conductivity of SCs.

The XPS technique was used to further investigate the precise chemical components of doped N and S atoms. The XPS spectra of samples distinctly showed the presence of C1s, O1s, N1s, and S2p peaks (Figure 4a). As shown in the XPS C1s spectra (Figure 4b), the main peak centered at 284.6 eV was attributed to C=C-C, whereas the peaks at 285.2, 288.2, and 290.8 eV belong to C-N/C-O, C=O, and π-π^*^, respectively. The surface content of N, S and elemental analysis are shown in ESI Appendix A. The surface content of N in N, S-DF-x (x = 1, 2, 3) carbon materials were 5.58, 9.02, and 5.64 at% and the surface content of S were 2.81, 4.23, and 1.47 at%, respectively. Nitrogen and sulfur content climbed up and then declined with the content of DFR, because of too much DFR being removed in the high-temperature pyrolysis process as a soft-template. In short, azo dye sludge wastes are potential N and S sources for N- and S-doped carbon materials. The N1s XPS spectrum of the N, S-DF- x (x = 1, 2, 3) showed four different N peaks, which include pyridinic-N (398.1 eV), pyrrolic-N (399.6 eV), graphitic-N (401.2 eV), and oxidized-N (403.5 eV) (Figure 4c, ESI Appendix A). According to XPS results (ESI Appendix A), N, S-DF-2 carbon materials possess absolute advantage in pyridinic-N (36.0%), pyrrolic-N (38.8%), and graphitic-N (19.7%) compared to N, S-DF-1, 3. Based on the research of nitrogen species, graphitic-N and pyridinic-N are significant components for catalytic activities [34,35]. Concretely, they are able to enhance diffusion-limited current density and onset potential, respectively. In addition, electrons transport and conductivity also profit from the N above two types. Two types of thiophenic-S (S 2p3/2 and S 2p1/2) and oxidized-S (S-O) are shown in Figure 4d, suggesting that S atom exist in defect sites and edges of carbon materials. Introducing S atoms could develop molecular kinetic and oxygen adsorption of ORR [36,37]. All in all, by introducing nitrogen and sulfur atoms, carbon atoms may facilitate the formation of adsorption sites for oxygen at ORR process.

The crystal structure and graphitization degree of N, S-DF-x were characterized by XRD and Raman, respectively. According to the XRD patterns (Figure 4e), the N, S-DF-2 exhibits two small diffraction peaks, being graphitic mark characteristic (002) and (100) of carbon materials [38,39]. Raman spectra of the prepared samples exhibit two distinct peaks in Figure 4f. Graphite degree also could derive from peak intensity of Raman spectroscopy. The peaks centered at 1345 cm^−1^ (namely D band) and 1580 cm^−1^ (G band), were involved in disorderly carbon and graphitic carbon, respectively [40]. The band intensity ratio of I_D_/I_G_ can serve as an indicator of the graphitic degree. According to the Raman spectra, the I_D_/I_G_ values of N, S-DF-x (x = 1, 2, 3) were calculated to be 0.86, 0.95, and 0.94, respectively, indicating the higher graphitization degree of N, S-DF-2 carbon materials. 

### 3.3. Electrochemical Performance of N, S-DF-x

The ORR performance of as-prepared carbon materials were evaluated by rotating disk electrode (RDE). Cyclic voltammogram (CV) curves of N, S-DF-2 electrocatalyst was performed in the N_2_- and O_2_-saturated 0.1 M KOH electrolyte. In Figure 5a, N, S-DF-2 electrocatalyst shows obvious cathodic peak at the range from 0 to 1.2 (V vs. RHE) in the O_2_-saturated KOH solution but did not appear in the N_2_-satureted condition. It indicates that the N, S-DF-2 electrocatalyst can improve ORR efficiently in the alkaline condition. Moreover, the reduction potential of N, S-DF-2 (Ep = 0.89 V) was higher than N, S-DF-1 (Ep = 0.81 V), N, S-DF-3 (Ep = 0.82 V), and the peak intensity of N, S-DF-1, 3 was not obvious (ESI Appendix A). To further explore the ORR activity of N, S-DF-x (x = 1, 2, 3), we conducted a linear sweep voltammetry (LSV) test in O_2_-saturated 0.1 M KOH at an electrode rotating speed of 1600 rpm. In Figure 5b, N, S-DF-2 and 20% Pt/C had almost the same onset potential (0.98 V) and limiting current density (5.46 mA·cm^−2^). The half-wave potential of N, S-DF-2 was 0.82 V and 20% Pt/C is 0.84 V. Obviously, the limiting current density of N, S-DF-2 was superior to that of N, S-DF-1 and N, S-DF-3. Apparently, pore structure of carbon materials can affect ORR activity. N, S-DF-2 possessed larger micropore pore volume and specific surface area than N, S-DF-x (x = 1, 3), which signifies that more active sites are exposed. Additionally, N, S element-modified carbon atoms can enhance electrocatalysis by facilitating the formation of adsorption sites for oxygen [41,42].

Furthermore, to evaluate the ORR activity of N, S-DF-2, the electron transfer number was calculated by K–L equation. Figure 5c shows typical LSV curves of different rotating speeds in 0.1 M KOH electrolyte of O_2_-saturated. LSV curves reveal the fact that, with rotation rates from 400 to 2500 rpm, current density increased as well. The reasonable explanation is diffusion distance gets shorter in higher speed conditions [43]. The K–L plots at different potentials 0.3–0.8 V showed excellent linearity and the electron transfer numbers are shown in Figure 5d. The results indicated that the ORR on N, S-DF-2 conform to a four-electron dominated pathway and the prime product is H_2_O. RRDE experiment is another remarkable method to confirm a four-electron pathway, as shown in Figure 5e. N, S-DF-2 displayed much lower ring current density (0.08 mA·cm^−2^) and indicates the minor content of hydrogen peroxide oxidation. The inset of Figure 5e clearly shows the electron transfer number and the amount of the generated H_2_O_2_. Next, chronoamperometric response measurement was used to estimate catalytic stability of N, S-DF-2. As shown in Figure 5f, when time gets to 35,000 s, N, S-DF-2 catalyst retains current densities at 65.4%, but commercial 20% Pt/C catalyst only retains current densities at 43.9% compared with initial current density. In addition, the methanol tolerance capability was also a significant consideration in order to estimate electrocatalysts. During the ORR test, 2.5 mL of methanol was added into KOH electrolyte to investigate the resistance of N, S-DF-2 and commercial 20% Pt/C. ESI Appendix A proved that the current density curves of commercial 20% Pt/C dropped quickly, while the N, S-DF-2 catalyst could stay stable gradually. Thus, not only durability but also crossover resistance of the N, S-DF-2 catalyst was better than commercial 20% Pt/C, undoubtedly.

Galvanostatic charge–discharge (GCD), CV, and cycle stability tests were conducted by a three-electrode system. Figure 6a exhibits the CV curves of N, S-DF-2 at different scan rates. The curves kept stable shapes when scan rate reached 50 mV·s^−1^. Even at 100 mV·s^−1^ the shape of the curves had little change. Thus, N, S-DF-2 electrode materials possess excellent capacitance retention with high charge–discharge rates. Figure 6b represents GCD curves of N, S-DF-2. The GCD curve was highly symmetric, responding to electric double-layer capacitive performance. Current density increased with decrease in charge–discharge time; because electrolyte ion diffusion needs a long time to enter porosity, only lower current densities meet this condition [44,45]. The specific capacitance value of the N, S-DF-2 sample reached 230.5 F·g^−1^ at a current density of 1 A·g^−1^ using Equation (7). Concretely, the specific capacitances of N, S-DF-2 were 246.2, 200.4, 175.3, and 150.6 F·g^−1^, responding to current density of 0.5, 2, 5, and 10 A·g^−1^, respectively (Figure 6c). Moreover, we compared the specific capacitances of N, S-DF-2 and other N-doped carbon materials (ESI Appendix A). The high specific surface area and porosity aid ion and electron exchange in the electrolyte, charge storage, and sequentially enhance the capacitance [46,47]. Lastly, electrochemical stability of the N, S-DF-2 electrode was tested by continuous GCD for 2500 cycles, as seen in Figure 6d. Current density was 1.0 and 10 A·g^−1^ responding to capacitance retention 100% and 98%, respectively. The results revealed that N, S-DF-2 has a high electrochemical stability and can serve as electrode material for supercapacitor devices.

## 4. Conclusions

In this work, we proposed a facile, green, and effective soft-template method to synthesize N, S dual-doped carbon materials via utilizing dye sludge. DFR takes a dual role, both as flocculant and soft template. The carbonization process formed a hierarchical pore structure by adjusting the content of DFR. The optimal carbon material (N, S-DF-2) displays significant performance compared to 20% Pt/C in alkaline medium, at the aspect of onset potential (0.98 V), limiting the current density (5.46 mA·cm^−2^). Even durability and crossover resistance is better than 20% Pt/C. In addition, N, S-DF-2 exhibits satisfied activity in electrochemical energy storage performance (230.5 F·g^−1^ at 1 A·g^−1^) and super capacitance retention (nearly 98% at 10 A·g^−1^) after 2500 cycles. This work utilized a simple and efficient soft-template method for in situ synthesis of N, S dual-doped carbon materials, applying to ORR and supercapacitors, which will be beneficial for the reuse and recycling of sewage sludge, as well as the production of good quality energy conversion and storage materials. Thus, this method will obtain high value-added products from sludge wastes and achieve industrial scale-up production.

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
