# Peer review of "N, S Dual-Doped Carbon Derived from Dye Sludge by Using Polymeric Flocculant as Soft Template"

_nanomaterials, 2019, doi:10.3390/nano9070991_

Round 1
Reviewer 1 Report
This manuscript concerns the design, preparation, physico-chemical and functional characterization of a N,S dual-doped carbon material for possible application in energy conversion and storage. The subject of the manuscript is relevant and worth to be investigated. The article is generally well-conceived: the work aim is clearly explained in the introduction, experimental set up is described in details, data are correctly presented and interpreted. Relevant literature is cited and comparison with previous works is reported. I think that this manuscript could be published on Nanomaterials, once the authors will have revised it considering the following points.
- My main concern is that the text is full of misprints, grammar mistakes and awkward expressions. It has definitely to be amended under this point of view. At the end of this review, I only list SOME corrections for the abstract and the introduction. It was impossible to list those needed in the rest of the manuscript.
- In the abstract, “performances” are cited while possible applications are still not reported. This is a little confusing. The authors should specify for which possible application they have tested the performance; otherwise, they should better refer to them as “physico-chemical properties”
- I have some doubts on Eq. 1: are the authors sure that the denominator is Ci (and not Co)?
- Line 115: I suppose ZNCl2 was used in solution. This has to be specified.
- Subsection 2.4. Please specify suppliers (with all details) for all the instruments used; moreover, many more details of the experimental procedures (instrument set up and so on) should be given.
- Eq. 3: define JL; Eq 4: define V ( or it is probably the Greek nu as at line 146)
- I do not find where Fig. 4b is cited
- Line 267: indicate in the figure which peaks you refer to
- Line 293: the authors discuss slight fluctuations that are hardy visible in the figure. This point has to be better introduced.
- Line 314: the electron transfer numbers are already shown in Fig. 5d, so they should not be repeated here.
- The conclusion section concisely and clearly summarizes the results. This is OK, but this section should also show the advancements this work brings in the research field, eventually indicating possible directions of future research.
Minor points
Line 14: “and occupy many places” please explain what you mean
Line 16: delete “these”
Line 22: “show” should be “shows”
Line 24: “open” should be “opens”
Line 27: “as well as production” should be “as well as for the production”
Line 31: “have been” should be “are”
Line 33: “bind with” should be “bind to”
Line 33: “and freely into” should be “ and are released into”
Line 34: “dye” should be “dyes”
Line 34: “with aromatic” should be “ with aromatic backbone”
Line 35: “with undesirable color” should be “leading to undesirable color”
Line 36: delete “which are”
Line 38: “even” should be “and even”
Line 41: “takes issues” should be “raises issues”
Line 42: “Transform” should be “Transforming”
Line 44: “on account of it involves environmental and energy concern” could be “considering that it concerns environmental and energy aspects” or similar
Line 46: “more concerns” should be “attention”
Line 48: “hinder their commercialization” should be “find difficulties in commercialization”
Line 48: “costly” should be “expensiveness”
Line 53: “have” should be “has”
Line 55: “materials include” should be “materials, including”
Line 57: delete “were”
Line 58: “enhanced” should be “enhancing”
Line 63: “attracts more interests” should be “attracts large interest”
Line 63: “represent attractive” should be “represents an attractive”
Line 64: “they contain” should be “it contains”
Line 78: “sources” could be “strategies”
Author Response
This manuscript concerns the design, preparation, physico-chemical and functional characterization of a N,S dual-doped carbon material for possible application in energy conversion and storage. The subject of the manuscript is relevant and worth to be investigated. The article is generally well-conceived: the work aim is clearly explained in the introduction, experimental set up is described in details, data are correctly presented and interpreted. Relevant literature is cited and comparison with previous works is reported. I think that this manuscript could be published on Nanomaterials, once the authors will have revised it considering the following points.
- My main concern is that the text is full of misprints, grammar mistakes and awkward expressions. It has definitely to be amended under this point of view. At the end of this review, I only list SOME corrections for the abstract and the introduction. It was impossible to list those needed in the rest of the manuscript.
- In the abstract, “performances” are cited while possible applications are still not reported. This is a little confusing. The authors should specify for which possible application they have tested the performance; otherwise, they should better refer to them as “physico-chemical properties”
- I have some doubts on Eq. 1: are the authors sure that the denominator is Ci (and not Co)?
- Line 115: I suppose ZNCl2 was used in solution. This has to be specified.
- Subsection 2.4. Please specify suppliers (with all details) for all the instruments used; moreover, many more details of the experimental procedures (instrument set up and so on) should be given.
- Eq. 3: define JL; Eq 4: define V ( or it is probably the Greek nu as at line 146)
- I do not find where Fig. 4b is cited
- Line 267: indicate in the figure which peaks you refer to
- Line 293: the authors discuss slight fluctuations that are hardy visible in the figure. This point has to be better introduced.
- Line 314: the electron transfer numbers are already shown in Fig. 5d, so they should not be repeated here.
- The conclusion section concisely and clearly summarizes the results. This is OK, but this section should also show the advancements this work brings in the research field, eventually indicating possible directions of future research.
Minor points
Line 14: “and occupy many places” please explain what you mean
Line 16: delete “these”
Line 22: “show” should be “shows”
Line 24: “open” should be “opens”
Line 27: “as well as production” should be “as well as for the production”
Line 31: “have been” should be “are”
Line 33: “bind with” should be “bind to”
Line 33: “and freely into” should be “ and are released into”
Line 34: “dye” should be “dyes”
Line 34: “with aromatic” should be “ with aromatic backbone”
Line 35: “with undesirable color” should be “leading to undesirable color”
Line 36: delete “which are”
Line 38: “even” should be “and even”
Line 41: “takes issues” should be “raises issues”
Line 42: “Transform” should be “Transforming”
Line 44: “on account of it involves environmental and energy concern” could be “considering that it concerns environmental and energy aspects” or similar
Line 46: “more concerns” should be “attention”
Line 48: “hinder their commercialization” should be “find difficulties in commercialization”
Line 48: “costly” should be “expensiveness”
Line 53: “have” should be “has”
Line 55: “materials include” should be “materials, including”
Line 57: delete “were”
Line 58: “enhanced” should be “enhancing”
Line 63: “attracts more interests” should be “attracts large interest”
Line 63: “represent attractive” should be “represents an attractive”
Line 64: “they contain” should be “it contains”
Line 78: “sources” could be “strategies”
Submission Date
07 June 2019
Date of this review
14 Jun 2019 10:21:58

Reviewer 2 Report
This work deals with the use of a polymeric flocculant for the synthesis of N and S doped carbon derived from dry sludge. The idea is interesting and promotes valorization. The subject belongs to nanomaterials. The authors is well written and easy to follow. The characterization of the different materials is good
Some minor points:
It is not clear to me why the authors incorporate also zinc to the material. It is necessary for oxygen reaction and the production of peroxide?
Can the authors provide us the zpc of the different materials? This is crucial for some applications.
Author Response
This work deals with the use of a polymeric flocculant for the synthesis of N and S doped carbon derived from dry sludge. The idea is interesting and promotes valorization. The subject belongs to nanomaterials. The authors is well written and easy to follow. The characterization of the different materials is good
Some minor points:
It is not clear to me why the authors incorporate also zinc to the material. It is necessary for oxygen reaction and the production of peroxide?
Can the authors provide us the zpc of the different materials? This is crucial for some applications.

Reviewer 3 Report
the subject is very interesting, however the methodology should be verified using a real dye flocculation sludge.
Author Response
Comments and Suggestions for Authors
the subject is very interesting, however the methodology should be verified using a real dye flocculation sludge.
Submission Date07 June 2019
Date of this review24 Jun 2019 17:48:47
